# Research on Thermal Stability and Properties of Ca₃ZrSi₂O₉ as Potential T/EBC Materials

**Yangyang Pan [1,2], Bo Liang [1,*], Yaran Niu [2,*], Dijuan Han [2], Dongdong Liu [2] and Xuebin Zheng [2,*]**

1   State Key Laboratory of Metastable Materials Science and Technology, College of Materials Science and Engineering, Yanshan University, Qinhuangdao 066004, China; panyangyang4155@163.com
2   Key Laboratory of Inorganic Coating Materials CAS, Shanghai Institute of Ceramics, Chinese Academy of Sciences, Shanghai 200050, China; djhan@njust.edu.cn (D.H.); 13102106737@163.com (D.L.)
*   Correspondence: liangbo@ysu.edu.cn (B.L.); yrniu@mail.sic.ac.cn (Y.N.); xbzheng@mail.sic.ac.cn (X.Z.)

**Abstract:** In this study, a new coating material for thermal barrier coating (TBC) or environment barrier coating (EBC) application, Ca₃ZrSi₂O₉ (CZSO), was synthesized and prepared by atmospheric plasma spray (APS) technology. The evolution of the phases and microstructures of the coatings with different thermal-aged were characterized by XRD, XRF, EDS and SEM, respectively. The thermal stability was measured by TG-DTA and DSC. The mechanical and thermal properties, including Vickers hardness ($H_V$), fracture toughness ($K_{IC}$), thermal conductivity ($\kappa$) and coefficient of thermal expansion (CTE) were focused on. It was found that the as-sprayed CZSO coating contained amorphous phase. Crystalline transformation happened at 900–960 °C and no mass changes took place from room temperature (RT) to 1300 °C. The phenomena of microcrack self-healing and composition uniformity were observed during thermal aging. The $\kappa$ of coating was very low at about 0.57–0.80 W·m⁻¹·K⁻¹ in 200–1200 °C. The combined properties indicated that the CZSO coating might be a potential T/EBC material.

**Keywords:** Ca₃ZrSi₂O₉ coating; thermal stability; thermal property; mechanical property; atmospheric plasma spray



## 1. Introduction

With the purpose of improving the efficiency and extending the service life of air or land-based advanced gas turbines, the TBCs are necessary in hot components [1,2]. As TBCs materials, the primary property is thermal insulation capacity, that is, low thermal conductivity ($\kappa$). Some researchers considered that the interstice between oxygen vacancies was vital for lower $\kappa$, except the relevant phonon mean-free-path (MFP) [3]. Smaller phono MFP and larger spacing resulted in lower $\kappa$. For this reason, a nanocrystalline yttria partially stabilized zirconia (YSZ) coating was developed, and almost all nanocrystalline YSZ samples were lower $\kappa$ than single-crystal YSZ.

Some researchers have found that the presence of the amorphous phase can also reduce the $\kappa$ [4]. According to quantum theory, thermal conductivity can be calculated by Equation (1). It shows that the main factor affecting the thermal conductivity is the mean-free-path of phonons. Qiu et al. [5] studied the effect of different content SiO₂ (amorphous phase) additions on the $\kappa$ of h-BN-La₂O₃-Al₂O₃-SiO₂ coatings. They found that the $\kappa$ was about 5 W/m·K when it contained 30 vol.% SiO₂, which was much lower than that of the coating without SiO₂ addition. Fleig et al. [6] analyzed the influence of the high grain boundaries on the $\kappa$, by establishing a brick layer model based on the finite element method, getting a formula for the influence of amorphous phase on $\kappa$, as follows Equation (2):

$$\kappa = \frac{1}{3} \int C_p \rho v l_p \tag{1}$$

$$\frac{1}{\kappa_{poly}} = \frac{1}{\kappa_{YSZ}} + n \times R \tag{2}$$

Which $\kappa_{poly}$ is the $\kappa$ of polycrystalline YSZ, $\kappa_{YSZ}$ is the $\kappa$ of single crystal YSZ, n is the grain boundaries' average number of per unit length, R is the thermal resistance value of this. Combining parameter fitting and experiments results, the R value of amorphous YSZ was about $2.54 \times 10^{-6}$ m$^2\cdot$K/W [6,7], which was three orders higher than that of the polycrystalline YSZ ($5 \times 10^{-9}$ m$^2\cdot$K/W). It can be seen that amorphous phase had great influence on $\kappa$.

Not only the spacing between oxygen vacancies and the size of the crystal grains, but also the complexity of the lattice and the types of atom are important factors influencing the $\kappa$. Some researchers have found that the $\kappa$ value of the monoclinic $ZrO_2$ was about 8.1 W/m·K at RT [8,9], if doping $Y_2O_3$ into $ZrO_2$ (YSZ), the $\kappa$ would drastically reduce to about 2.0 W/m·K [10,11]. The $\kappa$ of some typical TBC materials is listed in Table 1 [12]. It can be seen that $Sm_2Zr_2O_7$, $Yb_2Zr_2O_7$ and $La_2Zr_2O_7$ have lower $\kappa$ values than YSZ [13,14]. What is more, the quaternary compounds, such as $(Sm_{1/2}Yb_{1/2})_2Zr_2O_7$ and $(La_{1/2}Yb_{1/2})_2Zr_2O_7$, have much lower $\kappa$ than the ternary compound [15]. In our previous work [16], we found that $Ca_3ZrSi_2O_9$ has not only multi-component but also complex crystal packet structure. Therefore, we expect that this multi-component material might possess very low $\kappa$ and be a potential TBC.

**Table 1.** Thermal conductively of different typical TBC materials.

| Temperature (°C) | Binary | Ternary | | | Quaternary | |
|---|---|---|---|---|---|---|
| | m-$ZrO_2$ | $Sm_2Zr_2O_7$ | $Yb_2Zr_2O_7$ | 6–8 wt.%$Y_2O_3$ + $ZrO_2$ | $(Sm_{1/2}Yb_{1/2})_2Zr_2O_7$ | $(La_{1/2}Yb_{1/2})_2Zr_2O_7$ |
| 300 °C | ≈8.1 | 1.95 | 1.88 | 1.75 | 1.65 | 1.29 |
| 500 °C | - | 1.71 | 1.78 | 1.61 | 1.59 | 1.45 |
| 700 °C | - | 1.58 | 1.65 | 1.55 | 1.49 | 1.41 |
| 900 °C | - | 1.57 | 1.53 | 1.51 | 1.46 | 1.39 |

In this study, the $Ca_3ZrSi_2O_9$ (named CZSO) coating was successfully prepared by the APS technology. The powder and coating's phase evolution and microstructure were characterized. The coating's thermal stability was analyzed by TG-DTA, XRD and SEM. Some thermal properties ($\alpha$, $\kappa$ and CTE) were investigated. Besides, the mechanical properties (hardness and brittleness) of the CZSO coating were investigated for thermal aging from 30 h to 200 h at 1100 °C. This research indicates that $Ca_3ZrSi_2O_9$ might be a potential T/EBCs material for high temperature applications.

## 2. Materials and Methods

### 2.1. Powder and Coating Preparation

The CZSO powder was prepared by solid-state reaction using $ZrO_2$, $SiO_2$ and $CaCO_3$ (99.99% purity, Qinhuangdao Yinuo Advanced Material Co., Ltd., Qinhuangdao, China) as raw materials. The mixture powders ($CaCO_3$:$SiO_2$:$ZrO_2$ = 3:2:1, molar ratio) was sintered at 1400 °C for 6 h [16]. Then, the powder after sintering was agglomerated by spray drying and sintered again at 1200 °C for 3 h to obtain powder that was suitable for atmosphere plasma spray. The details of spray-drying could be seen in our previous work [17]. The coating was deposited by an air plasma spray system (A-2000, Sulzer Metco AG, Winterthur, Switzerland) equipped with a F4-MB torch. The spraying parameters used for the APS process were listed in Table 2. There was a water-cooling system on the back of aluminum substrate (120 mm × 80 mm × 2 mm) during coating was preparation, and then the free-standing coating (1.5–2.0 mm thickness) was mechanically removed from the substrate for observation and measurements.

**Table 2.** Operating parameters used for air plasma spraying.

| Powder (KW) | Primary Ar (slm) | Secondary H$_2$ (slm) | Carrier Ar (slm) | Spray Distance (mm) | Feeding Rate (rpm) |
|---|---|---|---|---|---|
| 46 | 41 | 14 | 2 | 120 | 25 |

The Thermo Gravimetry and Differential Thermal Analysis (TG–DTA) (STA449C, Netzsch, Selb, Germany) and the Differential Scanning Calorimetry (DSC204F1, Netzsch, Germany) of the powder and coating were performed. Then, 100 mg powder and $4 \times 4 \times 4$ mm$^3$ coating were used for TG-DTA and DSC measurements. The thermal diffusivity ($\alpha$) of the free-standing coating for 200–1200 °C was measured by the laser flash method. The disk specimen with a size of 11.0 mm (diameter) $\times$ 0.8 mm (width) was coated with a thin graphite layer to prevent the laser beam from directly transmitting through the sample. The density ($\rho$) of the specimens was measured by Archimedes method. The heat capacity (Cp) of different temperature stages was calculated by the Neumann–Kopp rule by the values of CaO, SiO$_2$ and ZrO$_2$ [18,19]. The $\kappa$ was calculated by Equation (3) [20]. The CTE of the coating was measured by using a high-temperature dilatometer (TMA403F3, Netzsch, Selb, Germany).

$$\kappa = \alpha \times Cp \times \rho \tag{3}$$

*2.2. Thermal-Aging*

Thermal-aging tests were conducted using an electric furnace (JXR1200-30, Shanghai Junke Co., Ltd., Shanghai, China). The samples were treated at 1100 °C for 3–200 h to investigate the phases and microstructure evolution. The compositions and contents of the powder or coating were performed by X-ray diffraction (XRD, RAX-10, Rigaku, Tokyo, Japan) and X-Ray fluorescence spectrometer (XRF, AXIOS, Panalytical, Almelo, Netherlands), respectively. The morphologies of the powder and the coating (surface and cross-section) were characterized by the Scanning Electron Microscopy (SEM, Magellan 400, FEI, Hillsboro, OR, USA). The coating porosity was evaluated by an image analysis software. Then, 5–8 back-scattered electron SEM (BSE-SEM) images, with 1000$\times$ magnification of the cross sections, were randomly taken for each specimen for image analysis.

The particle size distribution was performed by laser diffraction (Baite Instruments Ltd., Dandong, China). The hardness (Hv$_{4.9N}$) was measured by Vickers indentation (Shanghai Taiming Co., Ltd., Shanghai, China) with a load of 4.9 N. There were 10 points to be measured at random locations for every coating cross-section. The Evans–Wilshaw model was used to calculate the K$_{IC}$, which was related to c/a. The K$_{IC}$ values can be calculated by Equation (4). [21,22]. It is worth noting that it is well accepted that Equation(3) is effective when c/a is in 0.6–0.45 range.

$$K_{IC} = 0.079 \times (P \cdot a^{-3/2}) \times \log (4.5 \times a \times c^{-1}) \tag{4}$$

Which a is the half length of indentation diagonal, c is the length from the center of the indentation to the crack tip.

### 3. Results and Discussion

*3.1. Thermal–Physical and Microstructure Characterization of CZSO Powder*

The TG-DTA curves of the mixed powders with CaCO$_3$, SiO$_2$ and ZrO$_2$ (with a molar ratio of 3:2:1) are shown in Figure 1a. It was observed that the mass loss of the powders happened at about 600–830 °C, which was about 24%, and there was an endothermic peak observed. It might be that CaCO$_3$ decomposed to produce CaO with CO$_2$ (gas) and the CaO would be involved in the following reactions. Two exothermic peaks were found at 1340 °C and 1450 °C, respectively. It meant that the solid phase reaction happened in this temperature range. Figure 1b displays the XRD pattern of the CZSO powder after 6 h of solid phase sintering at 1400 °C. It was observed that the sintered powder was

mainly composed of the $Ca_3ZrSi_2O_9$ phase, mixed with a small amount of the $SiO_2$ and CaO phases.

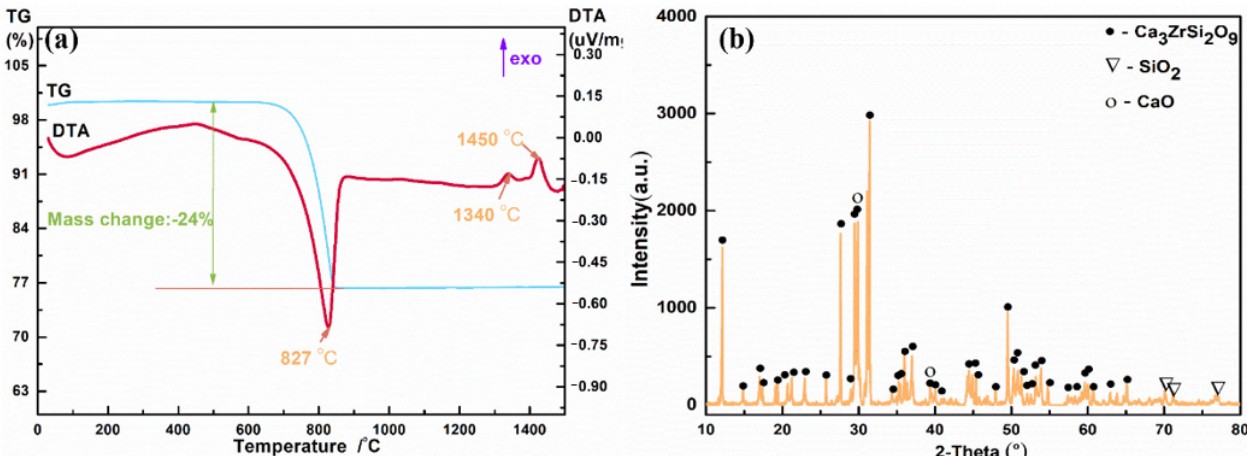

**Figure 1.** TG-DTA curve (**a**) of mixed powders before sintering ($CaCO_3$:$SiO_2$:$ZrO_2$ = 3:2:1 molar ratio) and XRD pattern (**b**) of $Ca_3ZrSi_2O_9$ powder after sintering.

The XRD pattern, surface morphologies and particle size distribution of the CZSO powder are presented in Figure 2. It can be seen that the phases of the powder almost had no change by comparing before and after spray-drying, and the powder has spherical morphologies (Figure 2b). The particle size of the powder was uniform and concentrated, whose D10, D50, D90 were about 9.5, 36.6 and 72.0 μm, respectively. During the plasma spray process, the spherical powder can ensure a sufficient powder feeding rate, and the uniform size of the powder can ensure a smooth process. According to the properties of the powder, the flow rate of Ar and $H_2$, the spraying distance and the powder feeding rate were adjusted to obtain excellent coating.

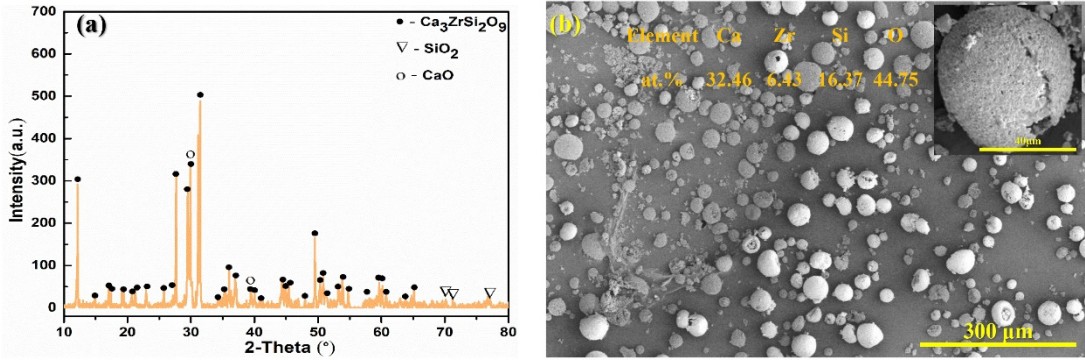

**Figure 2.** XRD pattern (**a**) and SEM morphologies (**b**) of CZSO powder after spray-drying.

### 3.2. Phase and Microstructure Characterization of Coating

The XRD pattern, surface morphologies and EDS results of the as-sprayed coating are presented in Figure 3. It was worth noting that the XRD of coating and powder should be measured at the same condition. By comparing, the XRD pattern of the as-sprayed CZSO coating was only a broad diffraction peak [23,24] and no $Ca_3ZrSi_2O_9$, $SiO_2$ and CaO diffraction peaks appearing, which was considered an amorphous structure (Figure 3a). This phenomenon was due to rapid cooling after the deposition process, and the atoms hardly diffuse to form an amorphous structure. Song et al. [25] also obtained the $Al_2O_3$-40 vol.% 8YSZ as-sprayed coatings with different contents of the amorphous structure by

changing the primary plasma gas during the APS process. It can be found that the coating was mostly built up by well-flattened splats, and there were also some regions with partly melted splats and splash debris. Some microcracks were found (Figure 3b,c), which could be related to the release of accumulated thermal stresses. The EDS results (the results in Figure 3c) proved that the contents of the four elements have not changed significantly, indicating that the CZSO was stable.

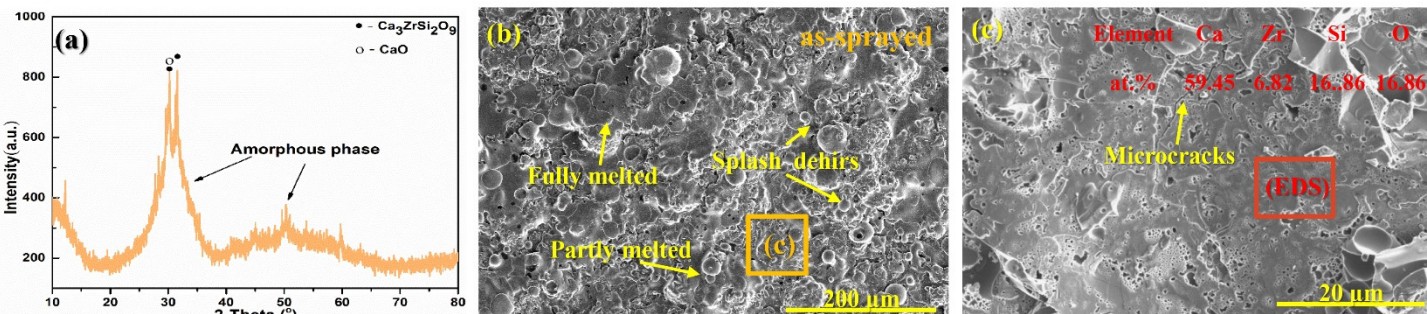

**Figure 3.** XRD pattern (**a**), surface morphologies (**b,c**) and EDS results of as-sprayed CZSO coating.

With the purpose of analyzing the composition evolution of the coating during deposition and thermal aging, the composition of the as-received powder and the coatings before and after thermal aging were characterized by XRF and are shown in Table 3. It can be found that the molar ratio of Si:Ca:Zr in the powder was about 35:49:16, which is similar to the as-sprayed and thermal aged coatings. It indicated that component of CZSO was stable during the plasma spray process.

**Table 3.** XRF results of the powder and coating before and after thermal aging.

| Content (at. %) | CZSO Powder | CZSO As-Sprayed | CZSO Coating 1100 °C/3 h |
|---|---|---|---|
| Si | 34.6 | 32.8 | 34.4 |
| Ca | 49.6 | 51.2 | 48.4 |
| Zr | 15.8 | 16.0 | 17.2 |

Figure 4 displays the micrographs of cross-section for the as-sprayed and 1100 °C/3 h aged CZSO coating. It was observed that the as-sprayed coating had a layered structure with some microcracks, and the color variation among splats was obvious, indicating compositional differences (Figure 4a,b). For the thermal aging coatings, the number of microcracks decreased and the layered structure gradually weakened and the color contrast also tended to be consistent, indicating that the composition distribution tended to be uniform (Figure 4c,d). The EDS results are summarized in Table 4. For the as-sprayed coating, it being shown that all points contained O, Ca, Zr and Si elements, excluding point 1. The molar ratio of point 1 was in agreement with $ZrO_2$. The results for points 2 and 3 justified the existence of $Ca_3ZrSi_2O_9$, and the atomic ratio of spot 4 and 5 was in agreement with $Ca_3ZrSi_2O_9 + ZrO_2$ and $Ca_3ZrSi_2O_9 + CaZrO_3$, respectively. For the thermal-aged coating, the molar ratios of points 8, 7, 9 and 6 were similar to points 1, 2, 3 and 5, respectively. Combining the EDS and SEM observations, it can be seen that the uniformity of elemental distribution was improved after thermal aging.

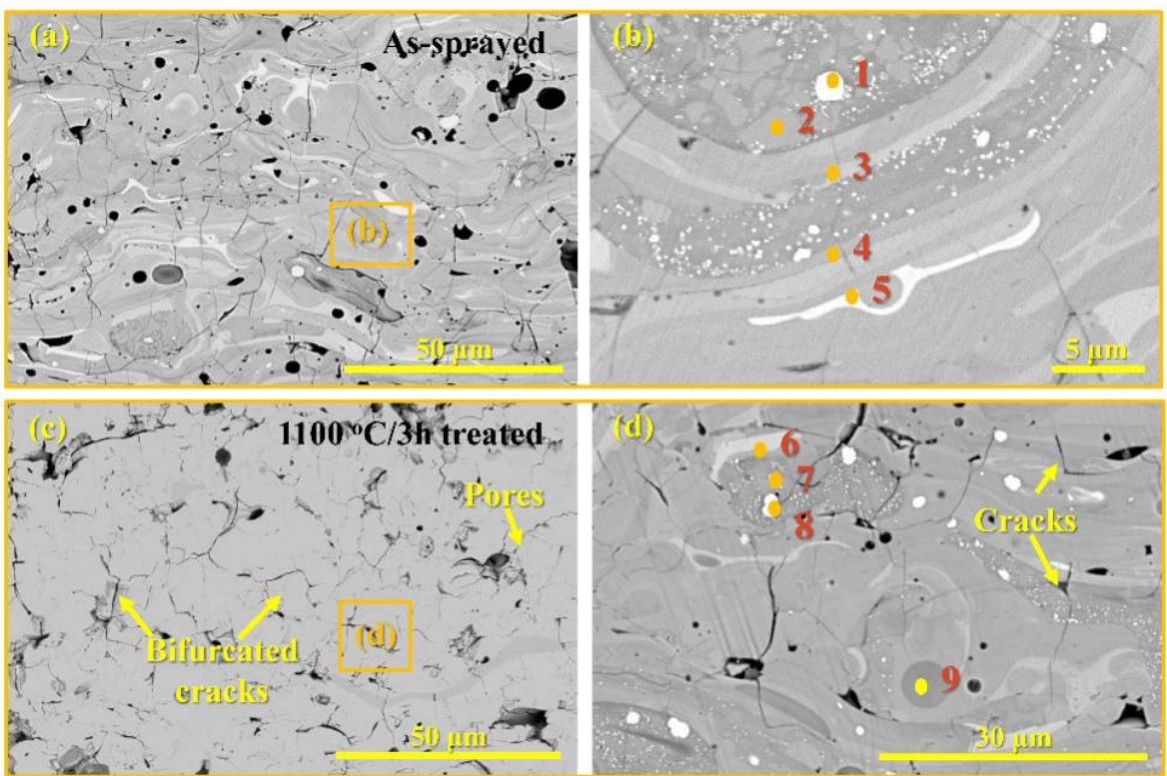

**Figure 4.** Cross-section morphologies of the as-sprayed (**a**,**b**) and thermal-aged (**c**,**d**) CZSO coatings.

**Table 4.** Element contents at each point in Figure 4.

|  | Point | O (at.%) | Si (at.%) | Ca (at.%) | Zr (at.%) |
|---|---|---|---|---|---|
| | $Ca_3ZrSi_2O_9$ | 60.00 | 13.33 | 20.00 | 6.67 |
| | 1 | 56.41 | 0 | 0.72 | 42.87 |
| | 2 | 52.04 | 16.37 | 22.84 | 8.75 |
| Figure 4b | 3 | 52.41 | 15.00 | 23.77 | 8.81 |
| | 4 | 52.04 | 13.15 | 18.96 | 15.85 |
| | 5 | 52.18 | 7.95 | 25.17 | 14.70 |
| | 6 | 42.65 | 7.08 | 37.93 | 12.36 |
| | 7 | 46.84 | 19.17 | 27.54 | 6.45 |
| Figure 4d | 8 | 55.89 | 3.83 | 1.14 | 39.13 |
| | 9 | 47.29 | 21.46 | 27.23 | 4.02 |

### 3.3. Thermal Stability of the Coatings

The TG-DTA and DSC curves of the as-sprayed CZSO coating are shown in Figure 5. There was almost no mass changing from RT to 1300 °C. Whereas, there were two sharp peaks at about 900 °C and 960 °C in the DTA curve, which might be related to the relaxation of amorphous phase. Two obvious exothermic peaks appeared at about 900 °C and 950 °C in the DSC curves (Figure 5b), which were similar to that of the DTA curve (Figure 5a). Indicating that the crystal transformation occurred at these two temperatures. The XRD pattern of the 1100 °C/3 h aged CZSO coating confirmed that the amorphous phase of the as-sprayed coating transformed into crystalline after 1100 °C/3 h thermal aging (Figure 5c).

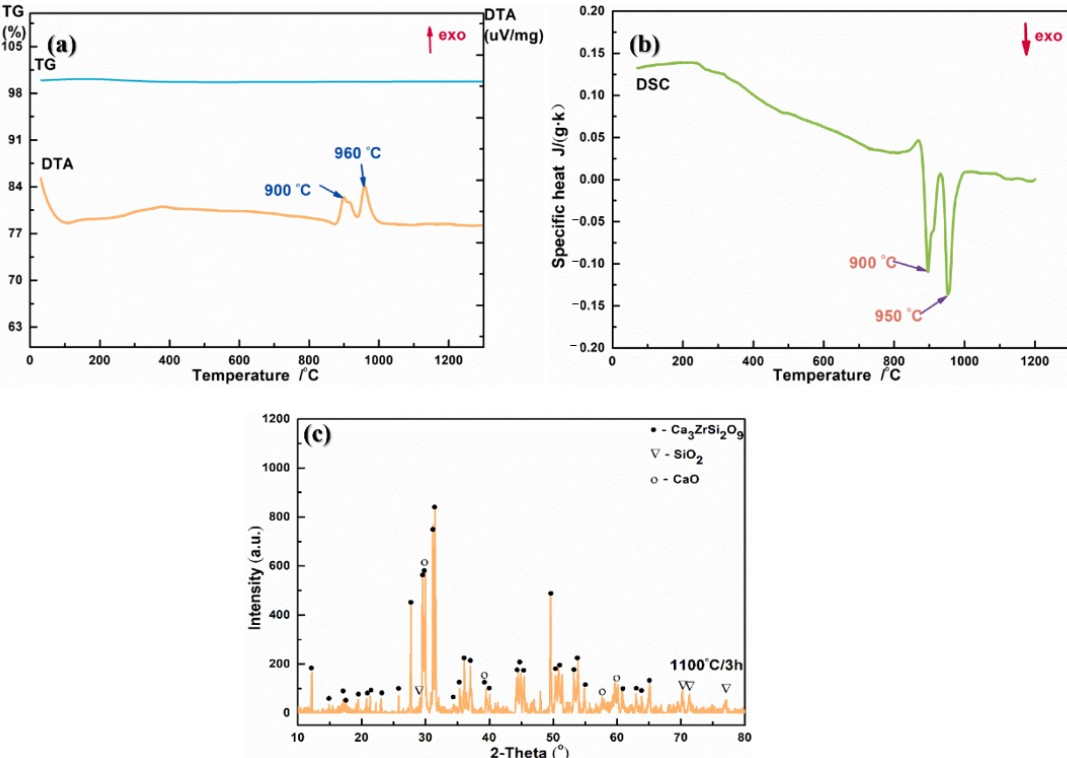

**Figure 5.** TG-DTA curves (**a**), DSC curve (**b**) of the as-sprayed and XRD pattern (**c**) of 1100 °C/3 h aged for CZSO coating.

The as-sprayed CZSO coating was aged at 1100 °C for 30–200 h, aiming to illustrate the long-term thermal stability. Figure 6 shows the XRD patterns of the thermal-aged coatings. Comparing the coatings after thermal aging for different time with the as-received powder, the mainly phase of the thermal-aged coatings was almost similar to that of the powder. This phenomenon indicated that the CZSO coating had good thermal stability at 1100 °C.

Figure 7 displays the micrographs of the cross-section with element mappings of the as-sprayed and 1100 °C aged CZSO coatings. It was observed that the layered structure gradually faded and the number of microcracks decreased. The distribution of Si and Zr elements gradually became uniform, and the Si-poor or Zr-rich regions gradually shrank over time. It means that the composition of the CZSO coating became more and more uniform, meanwhile, the CZSO coating has self-healing property. In addition, the porosities of the as-sprayed coating and 100 h and 200 h thermal aged coatings were 13%, 9% and 5%, respectively. Indicating that the cracks and pores decreased with thermal aging.

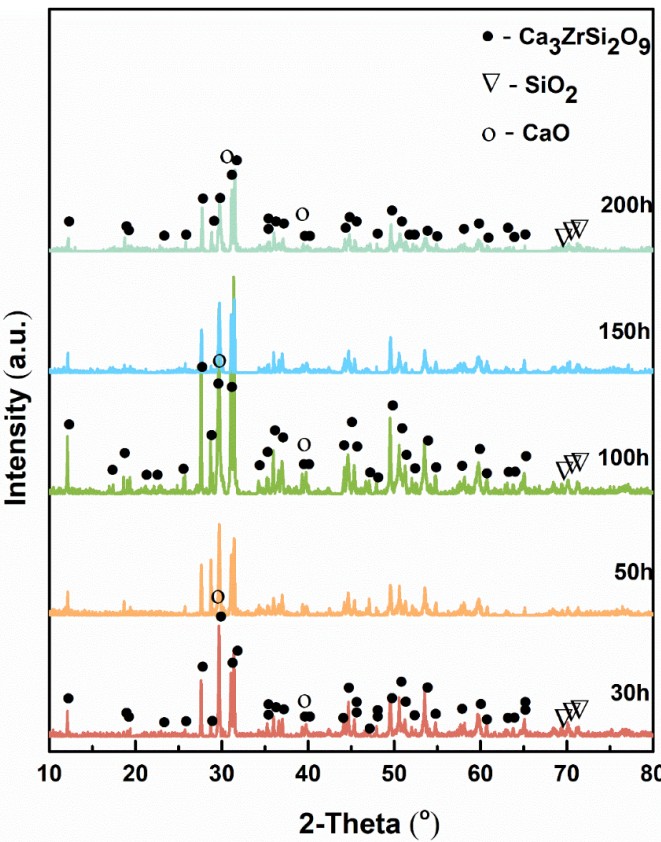

**Figure 6.** XRD patterns of the CZSO coating aged at 1100 °C for 30–200 h.

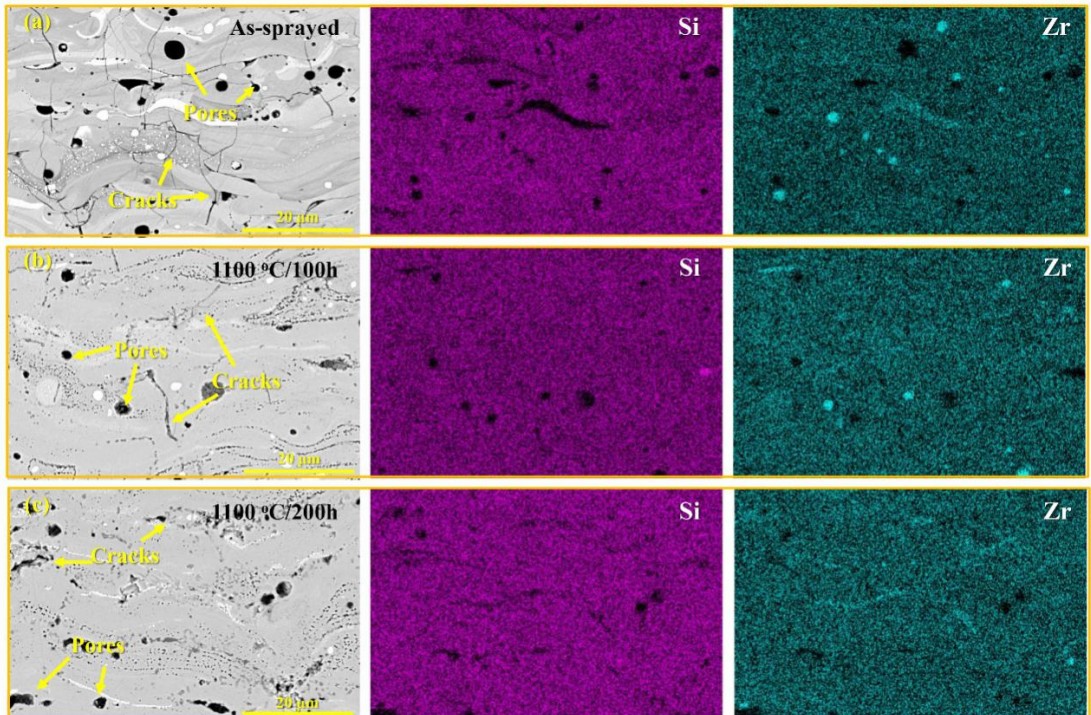

**Figure 7.** Cross-section morphologies with EDS results of as-sprayed (**a**), 100h (**b**) and 200h (**c**) of the CZSO coatings aged at 1100 °C.

With the purpose of enhancing the durability of T/EBCs materials, improving the tolerance of the coatings to the external damage was very important, which was closely related to their mechanical properties [26,27]. The $H_V$ values of CZSO coating are given in Table 5 and the indentation morphologies are shown in Figure 8. It was found that the hardness of the as-sprayed coating was about 1.73 ± 0.19 GPa; however. The value obviously raised to 2.86 ± 0.32 GPa after 1100 °C/30 h aging and did not change too much with the time extending. Comparing the indentation morphologies, it can be found that the as-sprayed coating had no cracks under 4.9 N load and short cracks appeared under 9.8 N load. For the thermal-aged coating, the length of cracks under 4.9 N became much longer after 30 h aging and did not change too much over time. This phenomenon indicated that the amorphous phase could help block crack propagation under external loads.

**Table 5.** Vickers hardness and fracture toughness of as-sprayed and 1100 °C aged CZSO coatings.

|  | As-Sprayed | 30 h | 50 h | 100 h | 150 h | 200 h |
|---|---|---|---|---|---|---|
| $H_{V4.9}$(GPa) | 1.73 ± 0.19 | 2.86 ± 0.32 | 2.69 ± 0.17 | 3.22 ± 0.18 | 2.83 ± 0.13 | 2.76 ± 0.44 |
| c/a | 0.77 (9.8N) | 0.83 (4.9N) | 1.30 (4.9N) | 1.04 (4.9N) | 1.09 (4.9N) | 1.22 (4.9N) |
| $K_{IC}$ (MPa·m$^{1/2}$) | 0.28 | 0.33 | 0.29 * | 0.31 * | 0.31 * | 0.29 * |

* means reference values.

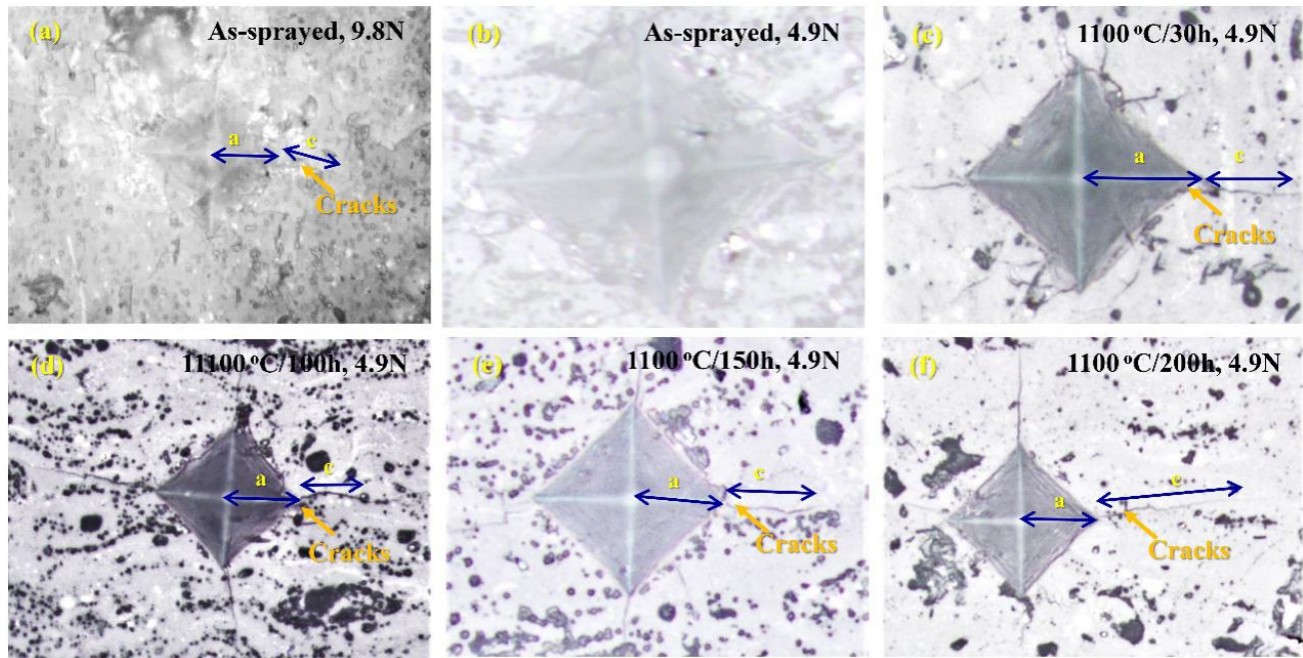

**Figure 8.** Indentation morphologies under different loads of as-sprayed (**a,b**), 30 h (**c**), 100 h (**d**), 150h (**e**) and 200 h (**f**) CZSO coating aged at 1100 °C.

Using value of the c/a can characterize the brittleness of the coating. The values of $K_{IC}$ are also summarized in Table 5. It was found that, for the as-sprayed coating, the values of c/a and $K_{IC}$ were about 0.77 and 0.28 MPa·m$^{1/2}$, respectively. For the 1100 °C/30 h aged coating, the values were about 0.83 and 0.33 MPa·m$^{1/2}$, respectively. With the aging time increasing, the c/a and $K_{IC}$ almost did not change.

### 3.4. Thermo-Physics Properties of As-Sprayed and 3 h Aged Coatings

The calculated $C_P$ curve of the CZSO is shown in Figure 9a. It is worth noting that there was a sudden decrease in 500 °C–600 °C range. This phenomenon can be related

to the sudden change in the $C_p$ value of $SiO_2$ [19]. It is well known that $SiO_2$ will change from $\alpha$-phase to $\beta$-phase at 574 °C, and the $C_p$ value will change from 75.395 J·mol$^{-1}$·K$^{-1}$ to 67.417 J·mol$^{-1}$·K$^{-1}$, as Figure 9a [19]. Figure 9b shows the $\alpha$ curves of the coatings in 25–1200 °C range. For the as-sprayed coating, it was found that the $\alpha$ value decreased with the temperature rising to 400 °C, and then increased a little during 400–600 °C, and then almost unchanged during 600–1000 °C, after that, increased to about 0.5 mm$^2$/s. For thermal aged coating, the $\alpha$ value tended to be 0.55–0.60 mm$^2$/s above 600 °C.

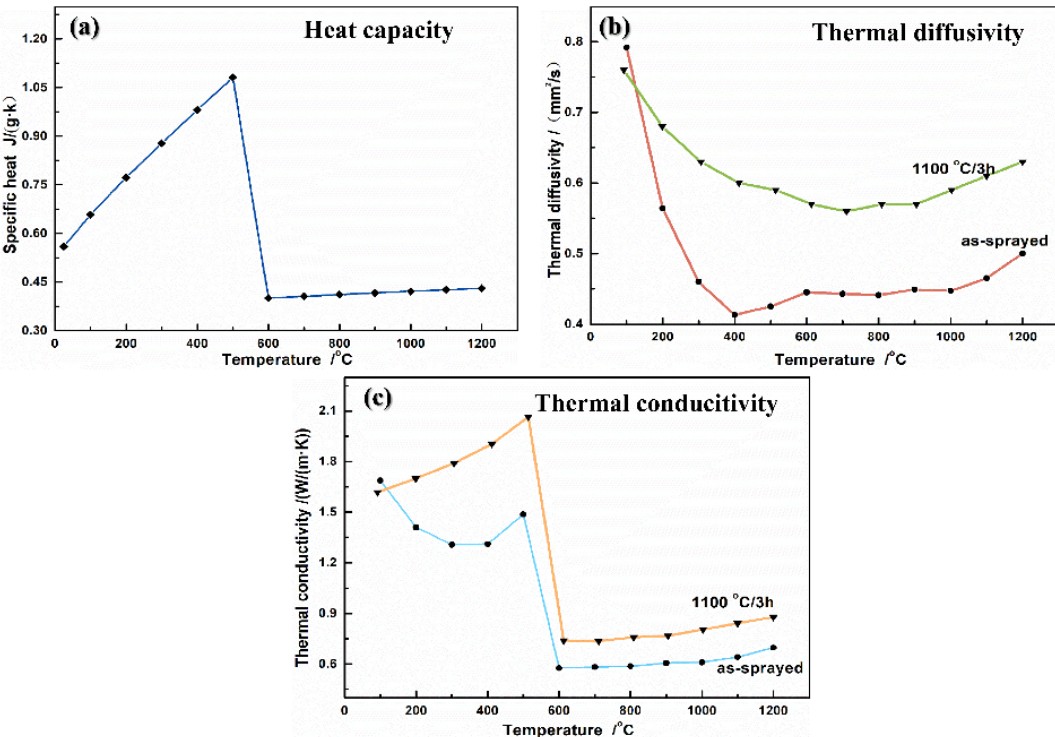

**Figure 9.** Heat capacity (**a**), thermal diffusivity (**b**) and thermal conductivity (**c**) of as-sprayed and 1100 °C/3 h aged CZSO coating.

Figure 9c shows the $\kappa$ curves from RT to 1200 °C. The used density of the CZSO coating was about 3.24 g·cm$^{-3}$. It can be found that a sharp decline appears at 500–600 °C, which was related to the influence of the calculated Cp value. For the as-sprayed coating, the $\kappa$ value was about 0.60 W/m·K at 600–1200 °C. For the thermal aged coating, the $\kappa$ values were a little higher, about 0.74–0.80 W/m·K at the temperature above 600 °C. It can be seen that the CZSO coating had lower $\kappa$ values and the thermal aging did not have much influence. However, from the reported of the rare earth zirconates at 900 °C, the $Yb_2Zr_2O_7$ was the lowest at about 1.53 W/m·K. The quaternary $(La_{1/2}Yb_{1/2})_2Zr_2O_7$ was 1.39 W/m·K. The thermal conductivity of CZSO materials was much lower than these. The crystal packet structure of $Ca_3ZrSi_2O_9$ is presented in Figure 10 [28]. It can be seen that there were two octahedrons of $CaO_6$, $ZrO_6$ and a tetrahedron of $SiO_4$, which was a relatively complex crystal packet structure. It can be thought that the complex crystal structure of $Ca_3ZrSi_2O_9$ contributes a lot to the relatively low $\kappa$ values.

It is well known that the mismatch of CTE between the top coating and substrates causes thermal stresses during thermal cycles, which is the main cause for spallation failure of T/EBC systems. Therefore, illuminating the thermal expansion behaviors of the CZSO coating is necessary, the results of which are shown in Figure 11. For the as-sprayed coating, the $\Delta L/L$ curve was non-linear for the first measurement. The shrinkage of the as-sprayed CZSO coating began at about 400 °C and became obvious at 850 °C and then the $\Delta L/L$ curve began to increase after 1000 °C. This phenomenon can be related to the crystallization of amorphous phase in the as-sprayed CZSO coating. The $\Delta L/L$

curve almost linearly increased at the second measurement. The CTE curve was obtained based on the second $\Delta L/L$ curve, which was $7.04$–$8.71 \times 10^{-6}\ \mathrm{K}^{-1}$ at 200–1200 °C. For the 1100 °C/3 h aged coating, the $\Delta L/L$ curve was linearly increased and the CTE was $6.89$–$8.70 \times 10^{-6} \cdot \mathrm{K}^{-1}$ at the same condition, which almost did not change. The CTE values of the CZSO coating ($7.04$–$8.71 \times 10^{-6} \cdot \mathrm{K}^{-1}$) were lower than YSZ coating ($9$–$10 \times 10^{-6} \cdot \mathrm{K}^{-1}$) and the superalloy ($16$–$17 \times 10^{-6} \cdot \mathrm{K}^{-1}$) [29]. However, the difference between CZSO coating with TiAl-alloy ($11$–$13 \times 10^{-6} \cdot \mathrm{K}^{-1}$) or SiC, C/SiC and SiC/SiC substrates ($4$–$6 \times 10^{-6} \cdot \mathrm{K}^{-1}$) [30] was smaller. It is supposed that CZSO coating might be suitable for these substrates as T/EBC materials. This phenomenon showed that the amorphous phase has great influence on the CTE. It was better to optimize the spray parameters to lower the content of amorphous phase.

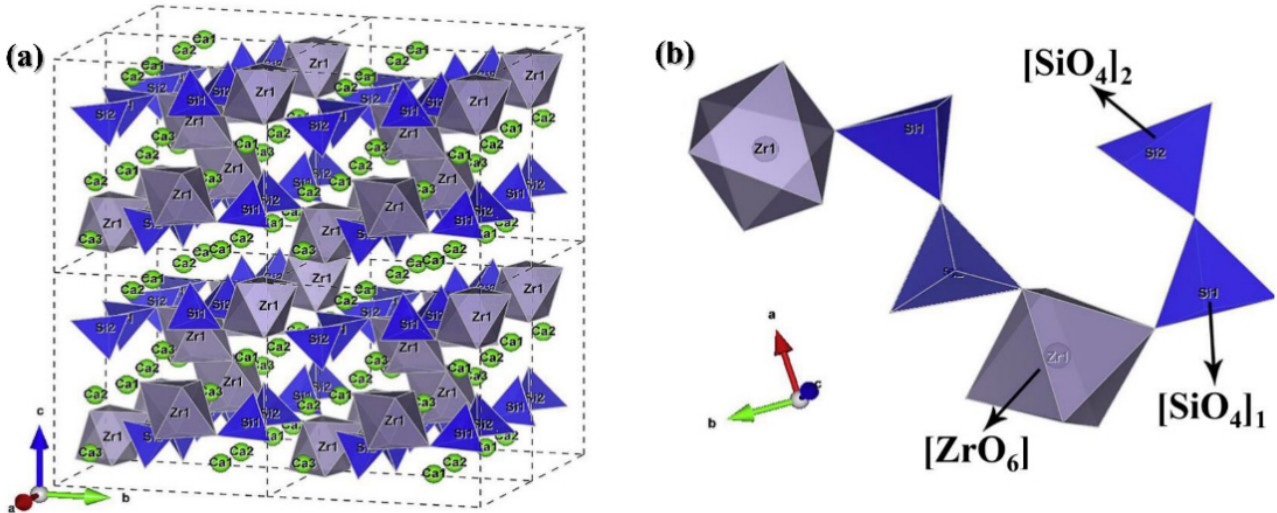

**Figure 10.** Crystal packet (**a**) and magnification (**b**) structure of $Ca_3ZrSi_2O_9$.

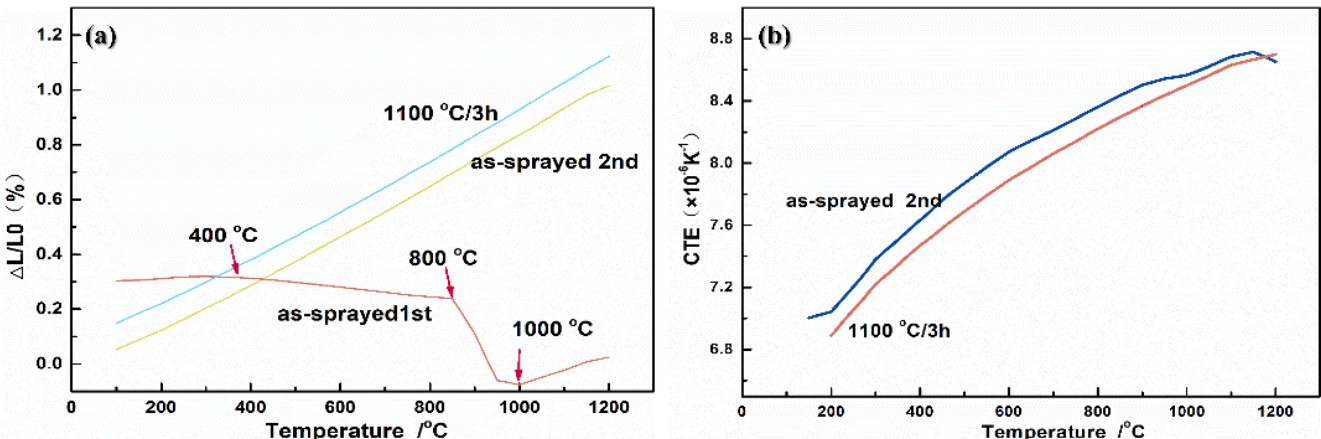

**Figure 11.** $\Delta L/L_0$ curves (**a**) and CTE curves (**b**) of as-sprayed and 1100 °C/3 h aged CZSO coating.

## 4. Conclusions

A new quaternary $Ca_3ZrSi_2O_9$ coating was prepared by the APS technique. The microstructure, thermal stability, and mechanical properties of the coating were characterized. The conclusions could be drawn as following:

- The as-sprayed CZSO coating presented lamellar microstructure and contained amorphous phase. The TG-DTA curves confirmed that no mass changed during RT-1300 °C and the crystalline phenomenon happened at about 900–960 °C. After aging at 1100 °C

for 30–200 h, the lamellar microstructure of the CZSO coating gradually disappeared, the composition became uniform, and the amount of microcracks decreased sharply.

- The $H_V$ and $K_{IC}$ of the as-sprayed CZSO coating were $1.73 \pm 0.2$ GPa and $0.28$ MPa·m$^{1/2}$, respectively, which were lower than that of the $1100\ ^\circ$C aged coatings. Besides, the as-sprayed coating had no crack under the 4.9 N load, however, there were cracks after aging at $1100\ ^\circ$C for 30–200 h.
- The thermal conductivity of both as-sprayed and thermal-aged CZSO coating was very low, about $0.57$–$0.80$ W·m$^{-1}$·K$^{-1}$, besides, the average CTE was $6.89$–$8.70 \times 10^{-6}$ K$^{-1}$ in 200–1200 $^\circ$C. The combined properties indicated that the $Ca_3ZrSi_2O_9$ coating might be a potential T/EBC material.
- To sum up the basic properties of the CZSO coating, it might be used on TiAl alloy, SiC, C/SiC or SiC/SiC substrates due to lower κ, CTE, and self-healing properties.

**Author Contributions:** Writing—original draft preparation, Y.P., D.H. and D.L.; writing—review and editing, B.L., Y.N. and X.Z.; All authors have read and agreed to the published version of the manuscript.

**Funding:** This work was supported by the Shanghai Technical Platform for testing on inorganic materials (19DZ2290700).

**Institutional Review Board Statement:** Not Applicable.

**Informed Consent Statement:** Not Applicable.

**Data Availability Statement:** Data sharing can be requested from the corresponding author.

**Conflicts of Interest:** The authors declare no conflict of interest.

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
