# Peer review of "Research on Thermal Stability and Properties of Ca3ZrSi2O9 as Potential T/EBC Materials"

_coatings, doi:10.3390/coatings11050583_

Round 1

Reviewer 1 Report

Please see the following comments and questions after review:-

Please check font type and sizing throughout the document, it seems to change continually.

Introduction

The references to amorphous phase content in polymers do not seem entirely appropriate when there are references for the influence of amorphous phase content on ceramics that would be more suitable to this article.

Table 1 - Monoclinic Zirconia is not considered a suitable or appropriate material for a TBC.

6-8wt% YSZ is the standard benchmark material across the industry, and it should be used for comparison. There are many references of thermal conductivity of this material across the temperature range specified.

Line 58-59 – discusses previous work on Ca3ZrSi2O9 but no reference is given?

Section 2.1

What was the feeding rate of the powder in grams per minute?

How thick were the coatings produced?

More information needs to be provided on how the APS spray powder was manufactured? How were the raw materials mixed, how was it agglomerated? How and when was it heat treated to achieve the required phase?

Section 2.2

More details need to be provided on the preparation of the samples and measurements performed.

How many samples were prepared for TG-DTA evaluation and how were they prepared.

How many samples were prepared for thermal diffusivity analysis? What were their dimensions and thickness? How were the samples prepared for the evaluation? Was a layer of graphite or gold applied to prevent transparency issues?

Section 3.2

The authors suggest that the coatings in this study present self-healing properties (line 210) due to the disappearance of cracks and lamellar porosity. However, the phenomena observed would also be consistent with sintering of an APS ceramic structure at high temperatures. This effect is also observed in YSZ and other TBC ceramics on long term ageing at higher temperatures and will result in loss of strain tolerance as well as increase in thermal conductivity. There are reference studies in thermal spray literature that document this.

Later in this section the measured fracture toughness is presented in table 5. Fracture toughness sems to be quite low for a thermal barrier coating. Please check literature references for other APS TBC fracture toughness’s for comparison and discuss the comparison.

Line 236 – section miss-numbered as 3.2 when it should be 3.3

Thermo-physical properties

Calculated Cp is utilised in this section, the reviewer is of the opinion that this approach is not accurate enough for a complex mixed oxide with highly heterogenous structure and inhomogeneous chemistry as presented here. Thermal spray coatings rarely behave as an ideal case in a calculation. The authors are advised to measure the specific heat capacity of the as-sprayed and heat treated coatings to better ascertain the coating thermal properties.

The thermal diffusivity values presented in figure 9 appear to be a reasonable measurement, though the thermal conductivity values would be under considerable doubt due to the calculated CP value utilised.

The density or porosity of the coatings is not presented or discussed at any point. This has an important influence on the thermal and mechanical properties of the coating.

The CTE of the coatings is presented in figure 11. The authors should comment on comparison between their materials measured CTE and that of other common TBC coatings. CTE mismatch between ceramic and metallic substrate can have an exceptionally large impact on the ability of the coating to survive a real-world TBC environment. As the authors point out, the first run shrinkage behaviour due to amorphous phase content may have a detrimental impact on coating behaviour.

Conclusions

The conclusions cover a summary of the results but lack some discussion on the broader suitability of the studied material for its intended use. How do the coatings investigated here compare with other TBC coatings currently utilised or investigated?

Some discussion should be made on future development steps or future work for this material/coating.  Currently the amorphous phase content and subsequent crystallisation seem to be a critical issue.

Reviewer 2 Report

I reviewed the article entitled “Research on thermal stability and properties of Ca3ZrSi2O9 as potential T/EBC materials” by Yangyang Pan et al. I believe that the work's objectives are clearly stated, and experiments were carried out based on them. But the article has a significant problem. The authors proposed the developed material for coatings, but they fail to do any experiment regarding coating. Moreover, the authors fail to defend/compare their results with the existing literature. The desired microstructure characters, hardness for coatings, is amorphous structure has any influence on coating? 

Reviewer 3 Report

  1. The work is interest but the concept of presenting and discussing the results is not clear. It is not clear why all presented aged samples are not also characterize in full, as the coating as sprayed or aged 3 hours at 1100 °C...
  2. In the entire manuscript there are two FONTS!
  3. Different techniques are used to characterize the obtained coating, but in fact, You have just one composition!
  4. The discussion of the results is not full and it is not in most of the cases related with the literature findings.
  5. Other comments You can find in pdf document.

Round 2

Reviewer 2 Report

As per the author's response, the article can be accepted.

Reviewer 3 Report

Dear authors the work can be accept in the present form.